ecology/behaviour/environmental science

climate change, Lepidoptera, line transects, long-term monitoring, light pollution, population decline

**Author for correspondence:**
Jens Rydell
e-mail: jens.rydell@telia.com

# Dramatic decline of northern bat *Eptesicus nilssonii* in Sweden over 30 years

Jens Rydell[1], Marcus Elfström[2,3], Johan Eklöf[4] and Sonia Sánchez-Navarro[1]

[1]Biology Department, Lund University, 223 62 Lund, Sweden
[2]EnviroPlanning AB, Lilla Bommen 5C, 411 04 Göteborg, Sweden
[3]Department of Natural Resources and Management, Norwegian University of Life Sciences, PO Box 5003, 1432 Ås, Norway
[4]Nattbakka Natur, 517 34 Bollebygd, Sweden

JR, 0000-0002-4930-6110

We monitored northern bat *Eptesicus nilssonii* (Keyserling & Blasius, 1839) acoustically along a 27 km road transect at weekly intervals in 1988, 1989 and 1990, and again in 2016 and 2017. The methodology of data collection and the transect were the same throughout, except that the insect-attracting mercury-vapour street-lights along parts of the road were replaced by sodium lights between the two survey periods. Counts along sections of the transect with and without street-lights were analysed separately. The frequency of bat encounters in unlit sections showed an average decline of 3.0% per year, corresponding to a reduction of 59% between 1988 and 2017. Sections with street-lights showed an 85% decline over the same period (6.3% per year). The decline represents a real reduction in the abundance of bats rather than an artefact of changed distribution of bats away from roads. Our study conforms with another long-term survey of the same species on the Baltic island of Gotland. Our results agree with predictions based on climate change models. They also indicate that the decline was caused directly by the disuse of the insect-attracting mercury-vapour street-lights, which may have resulted in lower availability of preferred prey (moths). In the 1980s, *E. nilssonii* was considered the most common bat in Sweden, but the subsequent decline would rather qualify it for vulnerable or endangered status in the national Red List of Threatened Species.

## 1. Introduction

Climate change models predict that species adapted to northern conditions will decline in the southern part of their range and the centre of distribution will retreat northwards, as the temperature

increases and potential competitors, parasites or pathogens extend their range from the south [1–3]. Among bats, the northern bat *Eptesicus nilssonii* (Keyserling & Blasius, 1839) is the northernmost species in the world [4], and is expected to respond to global warming in accordance with these predictions. Long-term monitoring studies of this species would, therefore, be of interest.

In the 1980s, *E. nilssonii* was considered the most common bat species in Sweden, and ever since then it has been classified as LC (least concern) at the national level, and no threats to the species have been identified [5]. According to IUCN [6], 'the species is widespread and abundant. No decline in population size has been detected, and there are no known widespread major threats'. Hence, the species is listed as LC globally as well. However, based on counts in several hundred bat hibernacula in continental Europe between 1993 and 2011, the population trend of *E. nilssonii* was reported to be 'uncertain' [7], thus not fully in agreement with the IUCN and the national evaluations.

There are very few long-term studies of *E. nilssonii* that can be used to evaluate its population trend, particularly in the northern part of its range, e.g. in Scandinavia. However, a repeated survey of *E. nilssonii* along roads on the Baltic island of Gotland (Sweden) between 1980 and 2014 indicated a decline of about 50%, which was attributed to habitat degradation [8]. Likewise, repeated winter counts of hibernating bats in Swedish mines over the same period also showed a significant decline, although the sample size was very small in this case [9]. If these surveys are representative, the real situation for *E. nilssonii* in Sweden differs from the national and IUCN evaluations.

The northern bat often feeds on insects attracted to lights, particularly street-lights [10–12], just as many other bats worldwide [13,14]. In the 1980s, most street-lights in Sweden and elsewhere were of the mercury-vapour (Hg) type, which included a UV-component that attracts insects such as noctuid moths (Lepidoptera, Noctuidae). These lights were later replaced by sodium (Na) lamps, which emit orange light, and then by white halogen or LED lights, all of which have relatively low attractiveness to insects [15,16]. The disuse of the insect-attracting Hg-lights resulted in changed distribution (lower patchiness and lower predictability, spatially and temporally) of insects and presumably decreasing availability of food for northern bats. This potentially important change in the *E. nilssonii* environment was not accounted for in any of the surveys mentioned above.

Trends in northern bat activity among years may reflect a real change in the abundance of the species. Alternatively, any apparent trend could be an artefact due to changed distribution of bats away from street-lights (and roads) in response to the disuse of insect-attracting Hg-lights, but with no real decline in the number of bats [17]. Hence, the aim of this study was (a) to analyse long-term trends in northern bat activity (1988–2017) in Sweden, and (b) to examine any effect of street-lights on the activity of *E. nilssonii*.

Based on specific scenarios, we made the following predictions:

1. If there has been no change in population size of the northern bat over the years, we would expect its activity to be unrelated to the year of monitoring.
2. If the population size has changed independently of street-lights, we would expect to see the same pattern in lit and unlit road sections.
3. If there was no change in overall northern bat population size, but reduced feeding activity in lit areas due to the change from insect-attracting street-lights to sodium lights, we would expect a reduction in bat activity along lit road sections but not in unlit sections.
4. Finally, if the population of northern bats has decreased over the years due to the change in street-lighting, we would expect a reduction in bat activity over the years, and also that this reduction is more pronounced in lit road sections than in unlit sections.

# 2. Material and methods

## 2.1. Field site

The study consists of repeated acoustic surveys of foraging northern bats along a 27 km road transect in southern Sweden over 30 years [10]. The transect was composed of eight sections of different lengths, four sections with street-lights alternating with four unlit ones. The lit sections were 8.2 km in total and the unlit sections were 19.3 km in total (table 1). The sections were located 250–300 m.a.s.l. in the province of Västra Götaland in southern Sweden (58° N, 13° E). A map showing the inventory route and its geographical location is shown in figure 1.

The habitat along the transect is typical for *E. nilssonii* in Sweden and as such also rather poor in other bats (four or five species). It consists of hemi-boreal forest, mostly pine *Pinus silvestris*, spruce *Picea abies*

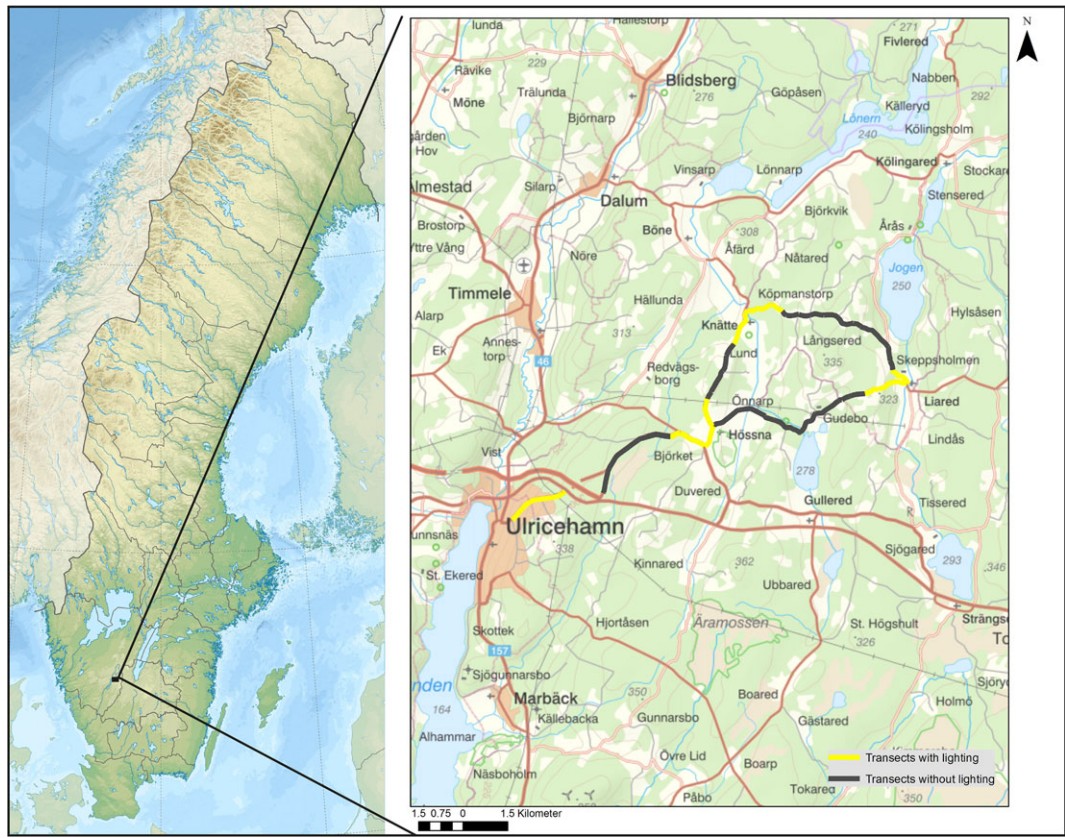

**Figure 1.** Location of the study area and the road transect in southern Sweden. Northern bat *Eptesicus nilssonii* was monitored with a bat detector in 1988–1990 and 2016–2017. Sections ($N = 8$) with (yellow) and without (black) street-lighting are indicated.

**Table 1.** Main characteristics of the eight sections comprising the 27 km line transect. *Lit and dark means the presence and absence, respectively, of street-lights. See also in figure 1.

| section no | length (km) | lit/dark* | habitats |
| --- | --- | --- | --- |
| 1 | 2.3 | lit | suburban |
| 2 | 3.7 | dark | forest |
| 3 | 3.1 | lit | village |
| 4 | 2.9 | dark | forest and farmland |
| 5 | 1.6 | lit | village |
| 6 | 6.0 | dark | forest |
| 7 | 1.4 | lit | village |
| 8 | 6.7 | dark | forest and farmland |
| total | 27.1 | | |

and birch *Betula* spp., and small farms with fields and pastures grazed by cattle, horse and sheep. The lit sections are in villages of various sizes and in one suburban area (table 1). The extension of built-up and lit areas remained the same over the 30 years of study and the number of street-lights also remained the same. Noticeable changes include a declining number of farm animals and replacement of many cows by sheep and horses. Also, the extent of mature forest has declined because of intensified forestry practices. Overall, the study area has nevertheless remained largely unchanged with respect to land use and associated habitats. The habitats and the changes that have taken place in the study area are generally representative for much of the country.

However, there are more subtle but perhaps more important changes that have taken place over the study period. For example, the mean annual temperature has increased by *ca* 0.8°C and the growing

season has become about a week longer [18]. At the same time, the soprano pipistrelle *Pipistrellus pygmaeus*, which was local and rather uncommon in the study area in the 1980s, is now abundant and may have replaced *E. nilssonii* as the most common bat species. Most importantly, the insect-attracting Hg-lights that prevailed along streets and roads in the 1980s, were replaced consistently by high-pressure sodium lights, and this shift presumably resulted in a lower availability of food for the bats, as indicated above.

## 2.2. Collection of field data

The data were collected along the transect at least once per week between August 1988 and October 1989 and also in March 1990, and then again between April 2016 and April 2017. The winter months (November to March) were included in the survey. Although winter feeding in this species seems unlikely at present, this may perhaps change with a warming climate in the future [19]. Nights with heavy mist, rain or snowfall were avoided, as well as nights with freezing temperatures, whenever possible, i.e. warmer and drier nights were chosen if available during the same week. Air temperature was measured at the start of the transect (in a suburban area), which took place *ca* 1 h after sunset, at the expected peak foraging activity of *E. nilssonii*. The route direction was alternated [10].

Driving at 30–50 km h$^{-1}$ with the detector microphone pointing obliquely upwards from the side window, passing bats were counted manually as they were heard through the headphones. The detector (D-960, Pettersson Elektronik AB, Uppsala, Sweden; www.batsound.se) was used in the heterodyne mode and tuned to 30 kHz, which corresponds to the energy maximum of the echolocation calls of *E. nilssonii* flying in open air [20]. The heterodyne system is particularly suitable in cases like this, because it automatically filters out noise such as that generated by the wheels touching the road, as well as the calls of other bat species. The northern bat is the only species in the study area using high-intensity echolocation centred at 30 kHz and it also has a highly characteristic pulse rhythm in most situations. It is easy to recognize acoustically and there is little risk of confusion with other species [11,20].

To facilitate the comparison among years, the method and equipment for data collection, including the detector and microphone, were the same during the whole study period and the same person (J.R.) made the counts on nearly all occasions. No stops were made along the transect and the driving speed was high enough to ensure that each bat could only be encountered once. We can, therefore, assume that the number of encounters with bats, as heard through the bat detector, was proportional to the number of bats active along the transect.

## 2.3. Seasonality

The data were collected throughout the year, but the seasonality in the study area is strong and the activity of bats varies substantially [11,21]. In spring (April and May), females are pregnant but nights are often cool and the feeding activity depends strongly on ambient temperature. Cool nights are spent in torpor. June and July (summer) correspond to late pregnancy and lactation, the ambient temperatures are higher and the females are active on nearly every night. In August to October (autumn), the activity declines from high to very low and is again strongly dependent on the air temperature [11,21]. The young are added to the population and the males are more active than earlier in the year. November to March is winter and the bats are in hibernation with little or no feeding activity.

Of particular importance for this study is that street-lights attract insects on which the bats feed in spring and autumn, when nights are dark. Around the summer solstice in late June the nights are not really dark at 57° N, and the lights attract few or no insects and bats. This change in contrast between lights and background drastically influences the distribution of insects and bats along the transect. Hence, bats are concentrated along lit road sections in spring and autumn but generally not in summer [10,11].

## 2.4. Statistical analysis

We analysed the temporal variation in bat activity, while controlling for ambient temperature, seasonality and the presence of street-lights. As there were few other apparent changes in environmental conditions among the years (see above), we assumed our bat monitoring to be a comparable index of temporal variation in bat activity and, thus, abundance.

All statistical analyses were carried out in R 3.5.1 [22]. We compared the bat activity among years by including ambient temperature, seasonality (spring, summer, autumn and winter), the presence/absence

**Table 2.** Model selection based on AICc values ($w_i$ = AICc weights) finding the most parsimonious generalized mixed-effect model when fitting northern bat (*Eptesicus nilssonii*) activity based on 1040 monitoring observations in Sweden during 1988, 1989, 1990, 2016 and 2017. Factor within parenthesis is treated as a random effect. (Obs, number of bat observations).

| candidate models | AICc | ΔAICc | $w_i$ |
|---|---|---|---|
| M1: Obs ~ year : lighting + season + temp + (transect section) | 2736.4 | 0.00 | 1.00 |
| M2: Obs ~ lighting + season + temp + (transect section) | 3124.8 | 388.40 | <0.00 |
| M3: Obs ~ (transect section) | 4259.2 | 1522.80 | <0.00 |

**Table 3.** Northern bat activity in relation to season (spring, summer, winter and with autumn as reference level), temperature and the interaction between year and lighting (present or absent), based on the most parsimonious GLMM (ΔAICc; $w_i$ = 1.00). Marginal variance for fixed effects (i.e. without random effects); $R^2 = 0.73$, and conditional variance (i.e. combining fixed and random effects); $R^2 = 0.82$. Values follow a Poisson distribution.

| variables | $\beta$ | s.e. | $z$-value | $p$-value (>|z|) |
|---|---|---|---|---|
| (intercept) | −1.395 | 0.265 | −5.264 | <0.00 |
| summer | 0.492 | 0.068 | 7.243 | <0.00 |
| winter | −1.819 | 0.244 | −7.462 | <0.00 |
| spring | −0.342 | 0.084 | −4.075 | <0.00 |
| temperature | 0.204 | 0.010 | 19.722 | <0.00 |
| year : lighting No | −0.030 | 0.004 | −7.479 | <0.00 |
| year : lighting Yes | −0.066 | 0.004 | −16.356 | <0.00 |

of street-lights and transect section, using generalized linear mixed-effect models (GLMM) with Poisson distribution from the package 'lme4' [23].

We constructed three *a priori* candidate GLMMs. The first model had year included together with ambient temperature, seasonality and street-light presence/absence. The second model had the same factors included except the exclusion of year. The third was an intercept-only model. Transect section was included as a random effect and other factors were treated as fixed effects in all models. We evaluated the most parsimonious GLMM to explain the variation in bat activity, based on Akaike's information criteria scores for small sample sizes (AICc) and AICc weights (AICcw) [24,25]. We calculated the variance explained as marginal and conditional r-squared ($R^2$) for the highest-ranked GLMM, based on the multi-model inference package 'MuMIn' [26].

We controlled for outliers with Cleveland dotplots, for multicollinearity with variance inflation factors [27], and for overdispersion by calculating a dispersion factor using the package 'blemco' [28].

## 3. Results

We made 130 drives along the transect (73 in 1988–1990 and 57 in 2016–2017) and had 1178 encounters with *E. nilssonii* (887 in 1988–1990 and 291 in 2016–2017). Hence, the dataset comprised 1040 monitoring events (N) derived from eight transect sections, where half of the monitoring events were made in the presence of street-lights, and half in its absence. The seasonal distribution was 272 monitoring events during spring, 144 during summer, 320 during autumn and 304 during winter. The annual distribution of monitoring events was 136 in 1988, 432 in 1989, 24 in 1990, 328 in 2016 and 120 in 2017.

The year was included in the GLMM with highest support to explain variation in bat activity (ΔAICc = 0.00, ΔAICcw = 1.00, $R^2_{marginal}$ = 0.73, table 2). Bat activity decreased both in the presence and absence of street-lights (table 3). In transect sections without street-lights, bat activity showed an average decrease of 3.0% (±0.4 s.e.) per year ($\beta_{GLMM}$ = −0.030 ± 0.004 s.e.; table 3 and figure 2). This corresponds to a 59% decline between 1988 and 2017. Bat activity in sections with street-lights decreased by 6.3% (±0.4 s.e.) per year ($\beta_{GLMM}$ = −0.066 ± 0.004 s.e.). Thus, without considering differences in light conditions among years, this corresponds to an 85% decline in bat encounters between 1988 and 2017 (figure 3).

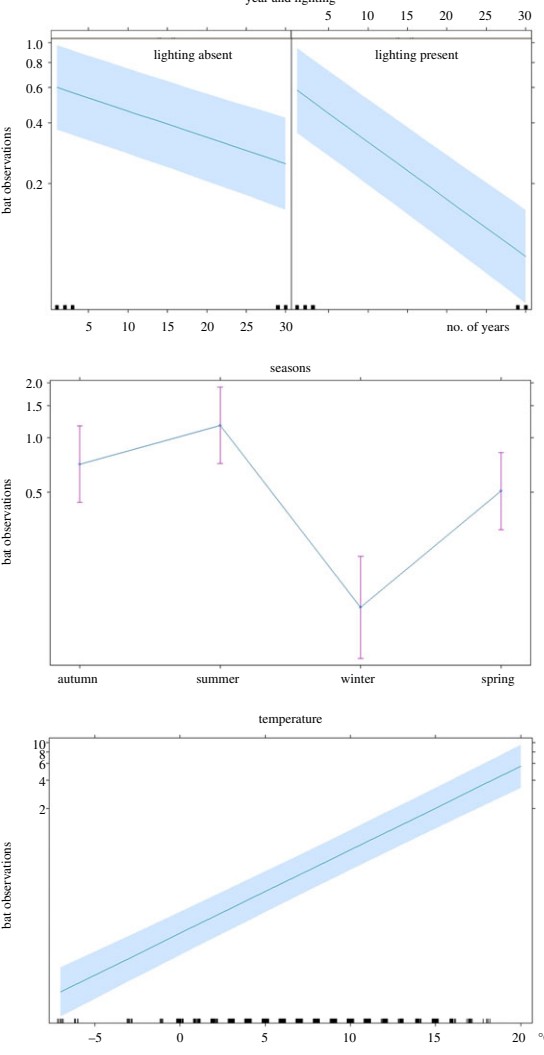

**Figure 2.** Separate effects and 95% confidence intervals on observations of northern bat Eptesicus nilssonii in relation to the presence of lighting and year, seasonality, and ambient temperature in eight different localities (transect sections) in Sweden between 1988 and 2017 based on the generalized mixed-effect model with the highest support ($w_i = 1.00$, $R^2_{marginal} = 0.73$).

Bat activity was positively correlated with ambient temperature ($\beta_{GLMM} = 0.204 \pm 0.010$ s.e.) corresponding to 22.6% ($\pm 1.0$ s.e.). The activity was 63.6% ($\pm 7.0$ s.e.) higher during summer ($\beta_{GLMM} = 0.492 \pm 0.068$ s.e.), 28.9% ($\pm 8.7$ s.e.) lower during spring ($\beta_{GLMM} = -0.342 \pm 0.084$ s.e.) and 83.8% ($\pm 27.6$ s.e.) lower during winter ($\beta_{GLMM} = -1.819 \pm 0.244$ s.e.), compared to the activity during the autumn. There were no bats observed along the transect during any of the winter periods (November to March) except for a single encounter during a warm weather spell on 13 March 2017.

## 4. Discussion

Our results indicate that *E. nilssonii* has declined dramatically in the study area over the 30 years since the start of the survey. A drastic decline was found both in the lit sections and those that remained unlit, while each of the different transect sections was controlled for. A similar decline of *E. nilssonii* was observed on the Baltic island of Gotland [8], although the authors of that report did not control for different lighting conditions during the study period. There is no indication that the situation is different in other areas or in neighbouring countries. Thus, northern bats seem to show a dramatic decline across Scandnavia, and we estimate this reduction to 59% over a 30-year period.

In a climate change scenario, we would expect the centre of distribution of *E. nilssonii* to shift northward, as the species declines in the south and perhaps increases in abundance in the north [1,2],

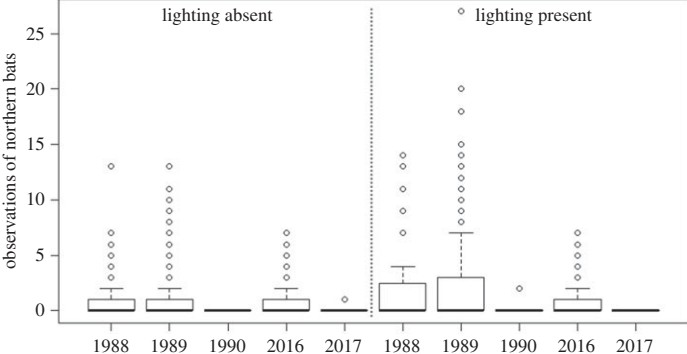

**Figure 3.** Boxplots (i.e. median, 1st and 3rd quartiles and range) of observations of northern bats *Eptesicus nilssonii* along a transect in south-central Sweden during the 30-year study period.

with or without any marked shift in the overall abundance. Our results seem to be in agreement with this scenario, but, unfortunately, information on changes in the species' distribution and abundance in the north is scarce and cannot be used to evaluate the hypothesis at present.

What could be the proximate causes for the observed decline? It was assumed [8] that the reduction recorded on Gotland was restricted to that island, and that the cause may be the recent destruction of forest through intensified forestry locally. However, as the present study shows an even stronger decline on the mainland, there are other explanations that we find more likely. For example, street-lighting is known to affect the distribution and feeding of *E. nilssonii* drastically [10,11]. Since all Hg-lights were replaced by Na-lights, which are much less attractive to insects (and bats [29]), during the course of the study, this seems a likely candidate explaining the overall trend.

Insect-attracting street-lights provide rich and highly profitable patches of large prey-items, notably noctuid moths (Lepidoptera, Noctuidae), at highly predictable sites and times [10–12]. Most importantly, the light suppresses the audition-based bat defence of eared moths such as noctuids and thereby makes them easy to catch, thus influencing the bats' prey selection and food intake rate [30,31]. It seems likely that *E. nilssonii* benefitted from this situation over several decades with increased food availability, presumably resulting in a population increase. However, the success of the bats feeding at street-lights may have occurred at the cost of the prey. If so, the Hg-vapour street-lights reversed the outcome of the bat–insect interactions locally, but how this may have affected the moth populations overall remains unknown. In any case, the winner in the evolutionary arms race suddenly became the loser [32].

Before World War II there were few street-lights in rural areas of Sweden. In 1947, Ryberg [33] stated that the most common bat at that time was the whiskered bat *Myotis mystacinus* (=*M. mystacinus* + *M. brandtii*), a 'species' which is very light-averse [34] and seldom or never feeds near lights [12]. Later, after the introduction of street-lights on a large scale, *E. nilssonii* was considered the most common species by Ahlén [20] and this view persisted until very recently [8]. However, this must be taken with great caution, because the two used very different methods to survey bats. While Ryberg [33] searched for colonies or roosting bats mostly using a torch, Ahlén [20] surveyed flying bats acoustically, using various bat-detectors.

The results agree with hypothesis 4 (see introduction), i.e. the change in street-lighting was the most important factor behind the observed decline. Nevertheless, we cannot rule out that other factors, which we did not measure, also could have been involved. For example, intensification of forestry practices and diminishing number of livestock could both have contributed to a general decline in insect abundance and hence to lower overall bat activity [8,35].

Could there be a competitive situation between *E. nilssonii* and pipistrelle populations expanding from the south, related to the global warming scenario? Indeed, *E. nilssonii* has been replaced as the most common bat in the southern part of Sweden by the soprano pipistrelle *Pipistrellus pygmaeus* during the course of this study, and, at the same time, two other pipistrelle species *P. pipistrellus* and *P. nathusii* have become established and are currently spreading northward [36]. Interspecific competition is notoriously hard to demonstrate in bats, but there seems to be a potential for interspecific dynamics among these species, and this may deserve further attention.

Our findings illustrate a generally interesting dilemma. Should we care about a species that once became common due to an anthropogenic driver, in this case, artificial lights, but now declines, perhaps to 'normal' levels, as the driver is removed? Outdoor lighting has been an important part of the habitat of the northern bat for many generations, just as houses and associated heating systems,

mines, root cellars and bird boxes, all providing potential roosting places. The same applies to e.g. livestock rearing and water eutrophication, which may benefit bats by increasing the abundance of certain Diptera [35,37]. Indeed, we believe that few if any Scandinavian bat species would qualify for conservation measures, if those that once have been boosted by anthropogenic activities cannot be considered. We do not see any principal difference between street-lights and e.g. water pollution or mines in this respect. It would be hard to distinguish anthropogenic factors that can be 'accepted' in a threatened species from those that cannot.

Clearly, our observed decline of 3.0% (±0.4 s.e.) per year, corresponding to 59% over 30 years (or four generations; the generation time is 6.6 years [6]), should be large enough to qualify *E. nilssonii* for vulnerable or even endangered status in the national Red List of Threatened Species. If the street-light hypothesis is correct, *E. nilssonii* may need attention in the rest of Europe as well, because Hg-vapour lights have been disused more or less throughout, and a similar decline may be expected. A 'recovery' of the population to the level before the change in street-lighting seems unlikely, because the removal of the insect attractant is permanent.

Conservation and monitoring of bats in Sweden focus on rare species and particularly species-rich localities, i.e. a tiny fraction of the country and its bats, while common species and ordinary habitats are largely ignored [38]. Management and conservation of wildlife in general also focus on rare species traditionally. The consequence of this philosophy is obvious from the result of this study. Major ecological events with wide-ranging implications for ecology and conservation, like the change in street-lighting, and the decline of the most abundant bat species by more than half, remained undetected (or was ignored) for more than a decade. Clearly, more and better information on the common bat species is badly needed.

Data accessibility. The dataset supporting this article has been uploaded on the Dryad Digital Repository: https://doi.org/10.5061/dryad.kwh70rz0b [39].
Authors' contribution. J.R. made the planning and collected the field data with assistance from J.E. and S.S.N. J.E. obtained funding. M.E. made the analysis and J.R. and M.E. wrote the draft with input from J.E. and S.S.N. All authors approved publication.
Competing interests. The authors have no competing interests.
Funding. We had no grants specifically for this project but enjoyed support from Harald and Gustaf Ekman's Foundation (2018-03-19) for studies on light pollution in general (to J.E.).
Acknowledgements. We acknowledge Martin Green, Johnny de Jong, Jeroen van der Kooij, Tore Michaelsen and Stefan Pettersson for comments on the manuscript. The study is dedicated to the memory of Jon Loman (†).

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
