## [Reviewer comments · Royal Society Open Science]

Review History

RSOS-191754.R0 (Original submission)

Review form: Reviewer 1

Is the manuscript scientifically sound in its present form?

Yes

Are the interpretations and conclusions justified by the results?

Yes

Is the language acceptable?

Yes

Do you have any ethical concerns with this paper?

No

Have you any concerns about statistical analyses in this paper?

No

Recommendation?

Accept with minor revision (please list in comments)

Comments to the Author(s)

Authors present very clearly their work documenting abundance of northern bats in a long-term study. They also propose multiple explanations for the decline and importantly suggest a revision of threatened status to reflect their findings. I only have minor suggests provided in a line by line format.

Line 14. Replace “by using a bat detector from a car” with “acoustically along”.

Line 19. Replace “kept separate” with “analyzed separately”.

Line 25. Replace “result” with “results”.

Line 26. Replace “It seem to agree qualitatively” with “This decline agrees”.

Line 27. Add in “may also be explained by” in front of “increased competition”.

Line 28. Delete “could perhaps be involved”.

Line 30. I prefer the use of “most common” in place of “commonest” but that may be personal preference only. See also Line 47, 245

Line 40. I suggest deleting “as well as common sense” because I don’t think this idea is widely known by non-scientists.

Line 65. Add in “(Hg)” because that is how you further refer to the lighting.

Line 73-80. Consider combing these two paragraphs and deleting the last sentence. It seems redundant.

Line 133-135. Please add an explanation of how you know a single bat wasn’t counted multiple times at a given stop along the transect, which would inflate your abundance measurements at some locations.

Review form: Reviewer 2

Is the manuscript scientifically sound in its present form?

Yes

Are the interpretations and conclusions justified by the results?

Yes

Is the language acceptable?

Yes

Do you have any ethical concerns with this paper?

No

Have you any concerns about statistical analyses in this paper?

No

Recommendation?

Accept with minor revision (please list in comments)

Comments to the Author(s)

The authors provide a very interesting analysis on activity trends of Northern bats in Sweden over a long time frame, illustrating an alarming negative trend. Their results indicate that such decline is mostly due to the change in the type of street lights, from Hg to Na, with a corresponding decrease in insect attraction and, thus, availability to bats.

Even though the decline found by the authors is probably genuine, as indicated by the decrease in bat activity also in non-lit areas, I am not completely convinced by the logical flow of the manuscript. I think that the authors should provide a list of hypotheses and associated

predictions according to different scenarios (e.g. "If there is no decline at all, we expect...; in case the change in light type produced a change in insect availability we expect). For example, if the change in activity they found was only due to the change in light type, I would expect an increase in unlit areas (or no changes at all). Thus, the general change suggests, in my opinion, that other factors may also play a role in this decline.

Also, even though I clearly understand the alarming tones of the authors when recording such a decline in the target species, I would also discuss the fact that the previous abundance of the species may have been boosted by artificial lights, i.e. by an anthropogenic driver, and that this may have had detrimental effects on other species in the past. My underlying question is: should we worry if a species that became common due to light pollution goes back to lower abundances when this pollution is reduced? I do believe such context should be clearly discussed in such a manuscript.

As a last comment, I would not push too much on the topic of interspecific competition, as this is very difficult to demonstrate in bats, and as such I would just mention that this is a potential dynamic occurring at lights, and that it deserves further attention.

Decision letter (RSOS-191754.R0)

22-Nov-2019

Dear Dr Rydell,

On behalf of the Editors, I am pleased to inform you that your Manuscript RSOS-191754 entitled "Dramatic decline of northern bat *Eptesicus nilssonii* in Sweden over 30 years" has been accepted for publication in Royal Society Open Science subject to minor revision in accordance with the referee suggestions. Please find the referees' comments at the end of this email.

The reviewers and handling editors have recommended publication, but also suggest some minor revisions to your manuscript. Therefore, I invite you to respond to the comments and revise your manuscript.

- Ethics statement

- Data accessibility

If you wish to submit your supporting data or code to Dryad (<http://datadryad.org/>), or modify your current submission to dryad, please use the following link:
<http://datadryad.org/submit?journalID=RSOS&manu=RSOS-191754>

- **Competing interests**

- **Authors' contributions**

- **Acknowledgements**

- **Funding statement**

Because the schedule for publication is very tight, it is a condition of publication that you submit the revised version of your manuscript before 01-Dec-2019. Please note that the revision deadline will expire at 00.00am on this date. If you do not think you will be able to meet this date please let me know immediately.

If your manuscript is newly submitted and subsequently accepted for publication, you will be asked to pay the article processing charge, unless you request a waiver and this is approved by Royal Society Publishing. You can find out more about the charges at <https://royalsocietypublishing.org/rsos/charges>. Should you have any queries, please contact openscience@royalsociety.org.

Kind regards,
Lianne Parkhouse
Editorial Coordinator
Royal Society Open Science
openscience@royalsociety.org

on behalf of Dr Punidan Jeyasingh (Associate Editor) and Kevin Padian (Subject Editor)
openscience@royalsociety.org

Associate Editor Comments to Author (Dr Punidan Jeyasingh):

This manuscript reporting data on the activity of bats over the last few decades was reviewed by two experts. Both were highly enthusiastic about the work (so am I). They raised a few points that

the authors should address before this manuscript can be accepted. Specifically, I agree with the reviewer suggesting the authors furnish a list of alternative hypotheses to lend a bit more structure to the manuscript. I also beg the authors to revise the manuscript to acknowledge limitations of the study (i.e. potential for unmeasured parameters to underlie observed patterns), and discuss the implications of the results in a measured fashion. With much gratitude to the expert reviewers, I invite the authors to make these revisions.

Reviewer comments to Author:

Reviewer: 1

Comments to the Author(s)

Authors present very clearly their work documenting abundance of northern bats in a long-term study. They also propose multiple explanations for the decline and importantly suggest a revision of threatened status to reflect their findings. I only have minor suggests provided in a line by line format.

Line 14. Replace "by using a bat detector from a car" with "acoustically along".

Line 19. Replace "kept separate" with "analyzed separately".

Line 25. Replace "result" with "results".

Line 26. Replace "It seem to agree qualitatively" with "This decline agrees".

Line 27. Add in "may also be explained by" in front of "increased competition".

Line 28. Delete "could perhaps be involved".

Line 30. I prefer the use of "most common" in place of "commonest" but that may be personal preference only. See also Line 47, 245

Line 40. I suggest deleting "as well as common sense" because I don't think this idea is widely known by non-scientists.

Line 65. Add in "(Hg)" because that is how you further refer to the lighting.

Line 73-80. Consider combing these two paragraphs and deleting the last sentence. It seems redundant.

Line 133-135. Please add an explanation of how you know a single bat wasn't counted multiple times at a given stop along the transect, which would inflate your abundance measurements at some locations.

Reviewer: 2

Comments to the Author(s)

The authors provide a very interesting analysis on activity trends of Northern bats in Sweden over a long time frame, illustrating an alarming negative trend. Their results indicate that such decline is mostly due to the change in the type of street lights, from Hg to Na, with a corresponding decrease in insect attraction and, thus, availability to bats.

Even though the decline found by the authors is probably genuine, as indicated by the decrease in bat activity also in non-lit areas, I am not completely convinced by the logical flow of the manuscript. I think that the authors should provide a list of hypotheses and associated predictions according to different scenarios (e.g. "If there is no decline at all, we expect...; in case the change in light type produced a change in insect availability we expect ..."). For example, if the change in activity they found was only due to the change in light type, I would expect an increase in unlit areas (or no changes at all). Thus, the general change suggests, in my opinion, that other factors may also play a role in this decline.

Also, even though I clearly understand the alarming tones of the authors when recording such a decline in the target species, I would also discuss the fact that the previous abundance of the species may have been boosted by artificial lights, i.e. by an anthropogenic driver, and that this

may have had detrimental effects on other species in the past. My underlying question is: should we worry if a species that became common due to light pollution goes back to lower abundances when this pollution is reduced? I do believe such context should be clearly discussed in such a manuscript.

As a last comment, I would not push too much on the topic of interspecific competition, as this is very difficult to demonstrate in bats, and as such I would just mention that this is a potential dynamic occurring at lights, and that it deserves further attention.

Author's Response to Decision Letter for (RSOS-191754.R0)

See Appendices A & B.

Decision letter (RSOS-191754.R1)

08-Jan-2020

Dear Dr Rydell,

It is a pleasure to accept your manuscript entitled "Dramatic decline of northern bat *Eptesicus nilssonii* in Sweden over 30 years" in its current form for publication in Royal Society Open Science. The comments of the reviewer(s) who reviewed your manuscript are included at the foot of this letter.

Kind regards,
Andrew Dunn
Royal Society Open Science Editorial Office
Royal Society Open Science

on behalf of Dr Punidan Jeyasingh (Associate Editor) and Kevin Padian (Subject Editor)
openscience@royalsociety.org

Associate Editor Comments to Author (Dr Punidan Jeyasingh):

I thank the expert reviewers for constructive comments and the authors for their careful attention to the same. This version is much improved. I am happy to recommend this paper for publication. Congrats to the authors for completing this laborious and important study!

Appendix A

Response to reviewers

Thanks for constructive reviewing!

Reviewer 1.

Revisions made in response to reviewer 1 are highlighted in green.

Line 14. Done as suggested.

Line 19. Done as suggested.

Line 25. Done as suggested.

Line 26. Done as suggested.

Lines 27 and 28. These two lines (sentence about interspecific competition) was removed from the abstract (in response to the last comment from reviewer 2), so this comment is no longer applicable.

Line 30. Done as suggested.

Line 40. Done as suggested.

Line 65. Done as suggested.

Line 73-80. Done as suggested.

Line 133-135. Done as suggested.

Reviewer 2.

Revisions made in response to reviewer 2 are highlighted in red.

Second paragraph. We appreciate the comment regarding the logical flow. We have provided a list of hypotheses and associated predictions, as suggested. We have revised accordingly, specifically lines 77-87.

Third paragraph. We agree that a comment about the anthropogenic source of the increase and subsequent decline is interesting and important particularly in this case. We have written a new paragraph to fulfill this suggestion (lines 265-276).

Fourth paragraph. We also agree with the view on the complexity of interspecific competition dynamics, and have removed most of the competition section with its references. We revised as suggested in a new paragraph (lines 257-264).

Appendix B

Response to reviewers

Thanks for constructive reviewing!

Reviewer 1.

We revised as suggested. All changes are highlighted in **green**.

Reviewer 2.

a. We appreciate the comment regarding the logical flow. We have provided a list of hypotheses and associated predictions, as suggested. We have revised accordingly, specifically lines 80-90. Revisions made in response to reviewer 2 are highlighted in **red**.

b. We agree that a comment about the anthropogenic source of the increase and subsequent decline is interesting and important in this case. We have written a new paragraph to fulfill this suggestion (lines 268-282).

c. We agree on the complexity of interspecific competition dynamics, and have removed most of the competition section with its references. We revised accordingly (lines 265-267).